# Acetyl-CoA production by specific metabolites promotes cardiac repair after myocardial infarction via histone acetylation

**Ienglam Lei[1,2†], Shuo Tian[2†], Wenbin Gao[2†], Liu Liu[2], Yijing Guo[2], Paul Tang[2], Eugene Chen[2], Zhong Wang[2]***

[1]Faculty of Health Sciences, University of Macau, Taipa, China; [2]Department of Cardiac Surgery, University of Michigan-Ann Arbor, Ann Arbor, United States

**Abstract** Myocardial infarction (MI) is accompanied by severe energy deprivation and extensive epigenetic changes. However, how energy metabolism and chromatin modifications are interlinked during MI and heart repair has been poorly explored. Here, we examined the effect of different carbon sources that are involved in the major metabolic pathways of acetyl-CoA synthesis on myocardial infarction and found that elevation of acetyl-CoA by sodium octanoate (8C) significantly improved heart function in ischemia reperfusion (I/R) rats. Mechanistically, 8C reduced I/R injury by promoting histone acetylation which in turn activated the expression of antioxidant genes and inhibited cardiomyocyte (CM) apoptosis. Furthermore, we elucidated that 8C-promoted histone acetylation and heart repair were carried out by metabolic enzyme medium-chain acyl-CoA dehydrogenase (MCAD) and histone acetyltransferase Kat2a, suggesting that 8C dramatically improves cardiac function mainly through metabolic acetyl-CoA-mediated histone acetylation. Therefore, our study uncovers an interlinked metabolic/epigenetic network comprising 8C, acetyl-CoA, MCAD, and Kat2a to combat heart injury.

**\*For correspondence:**
zhongw@med.umich.edu

[†]These authors contributed equally to this work

**Competing interest:** The authors declare that no competing interests exist.

## Editor's evaluation

The authors present evidence for a novel role of acetyl CoA in response to different carbon sources and histone acetylation in the context of myocardial ischemia-reperfusion injury following myocardial infarction. The findings will add to the understanding of mechanisms underlying the pathophysiology of reperfusion injury and introduce novel potential targets for therapeutic intervention.

## Introduction

Energy is a fundamental requirement for all living organisms and its production typically requires fuels (metabolites) and oxygen. Heart function is sensitive to energy supply and provision. One key response to fuel changes is the epigenetic modifications using one carbon or two carbon moieties derived from metabolites to change chromatin structure by methylation and acetylation and regulate gene expression *Reid et al., 2017*. Notably, a bi-directional interplay between metabolism and epigenetic control has been proposed recently: metabolism directly regulates chromatin epigenetic state, whereas chromatin state defines gene control in response to metabolic status *Pietrocola et al., 2015*; *Shi and Tu, 2015*.

Myocardial infarction (MI), which blocks the supply of energy to infarcted area, is one of the leading causes of death in the world *Pagidipati and Gaziano, 2013*. Despite the severe complications of this

devastating disease *Ibáñez et al., 2015*, the therapies to reduce MI injury are still limited. MI is accompanied by severe energy deprivation and extensive epigenetic changes. Regulations in either energy metabolism or epigenetics are essential for heart function and pathogenesis *Keating and El-Osta, 2015*; *Maack and Murphy, 2017*, but how these two events are interlinked in the context of MI has been under-explored. Clearly, exploring an integrated metabolic and chromatin control in MI injury and heart repair and regeneration may provide novel therapies against heart injury.

Acetyl-CoA, an intermediary metabolite that serves as an energy-providing substrate, is also used by histone acetyltransferases (HATs) to transfer the acetyl-group to histone residues for histone acetylation *Cai et al., 2011*; *Wellen et al., 2009*. Manipulation of acetyl-CoA, either by intervention of synthetic enzymes, or nutrient source could alter histone acetylation in various cell types *Wellen et al., 2009*; *Sutendra et al., 2014*; *Zhao et al., 2016*; *McDonnell et al., 2016*. Importantly, acetyl-CoA induces metabolic adaptations through regulating histone acetylation in response to starvation or hypoxic conditions *Bulusu et al., 2017*; *Gao et al., 2016*. Moreover, myocardial ischemia reperfusion (I/R) injury causes dramatic metabolism changes and subsequent histone deacetylation. Inhibition of histone deacetylation activity protects heart function after I/R injury *Tian et al., 2019*; *Xie et al., 2014* and inhibits cardiac remodeling and heart failure *Jeong et al., 2018*. Thus, the intersection of acetyl-CoA-mediated metabolism and histone acetylation is very likely a novel hub for identifying targets for heart repair.

In this study, we examined whether metabolic manipulation of acetyl-CoA level could alter histone acetylation, which in turn promotes heart repair and protection after I/R injury. Our screen identified a nutrient, the medium chain fatty acid 8C, as an effective carbon source to stimulate acetyl-CoA production and histone acetylation and protect heart function after MI. Specifically, we showed that a single i.p. injection of 8C at the time of reperfusion significantly reduced the infarct size and improved cardiac function at both 24 hr and 4 weeks after MI. We found that 8C produced acetyl-CoA and rescued histone acetylation decrease both in vivo and in vitro after I/R injury. We further elucidated that the metabolic enzyme medium-chain acyl-CoA dehydrogenase (MCAD) and histone acetyltransferase Kat2a were key factors in transferring the acetyl moiety in 8C to acetyl groups in histone acetylation. Knockdown of either MCAD or Kat2a abolished 8C-mediated histone acetylation and subsequently oxidative stress reduction for cardiomyocyte protection after I/R injury. Thus, our study established an interlinked metabolic/epigenetic network that may provide new strategies to treat heart injuries.

## Results

### Administration of 8C at reperfusion elevated acetyl-CoA and improved short-term and long-term cardiac function after I/R

Myocardial infarction (MI) induces dramatic metabolic and epigenetic changes including decrease of acetyl-CoA synthesis and histone acetylation *Bodi et al., 2012*; *Granger et al., 2008*. We reasoned that swift synthesis of acetyl-CoA could rescue heart function by restoring histone acetylation. Based on the major metabolic pathways for acetyl-CoA synthesis (*Figure 1A*; *Pietrocola et al., 2015*), we examined the effect of sodium acetate (2C, 500 mg/kg), sodium pyruvate (3C, 500 mg/kg), sodium citrate (6C, 500 mg/kg), sodium octanoate (8C, 160 mg/kg), and sodium nonanoate (9C, 200 mg/kg) on heart function after I/R. Rats were intraperitoneally (i.p.) injected with these metabolites as well as saline for three days and another dose before I/R surgery (*Figure 1B*). At 24 hr after I/R, the infarct size was measured by Evans blue and tripheyltetrazolium chloride (TTC) staining at 24 hr after I/R. Evans blue staining indicated area at risk (AAR) and TTC staining indicated the infarct size (IS). The ratio of AAR/LV was similar in all groups (*Figure 1—figure supplement 1A-B*). Interestingly, these metabolites displayed distinct effects on reducing myocardial infarct size. Among the five metabolites examined, 2C, 3C, and 8C significantly reduced the IS/AAR ratio, whereas 6C and 9C treatment did not reduce IS/AAR ratio compared to saline-treated groups (*Figure 1C* and *Figure 1—figure supplement 1A-B*). These results indicated a potential beneficial effect of replenishment of acetyl-CoA with specific metabolites for heart repair after I/R.

As 8C administration resulted in the most dramatic protection among all the metabolites tested after I/R (*Figure 1C*), we focused on investigating the role of 8C on heart protection in a more clinically relevant setting. 8C (160 mg/kg) or saline were i.p. injected only once at the time of reperfusion, which

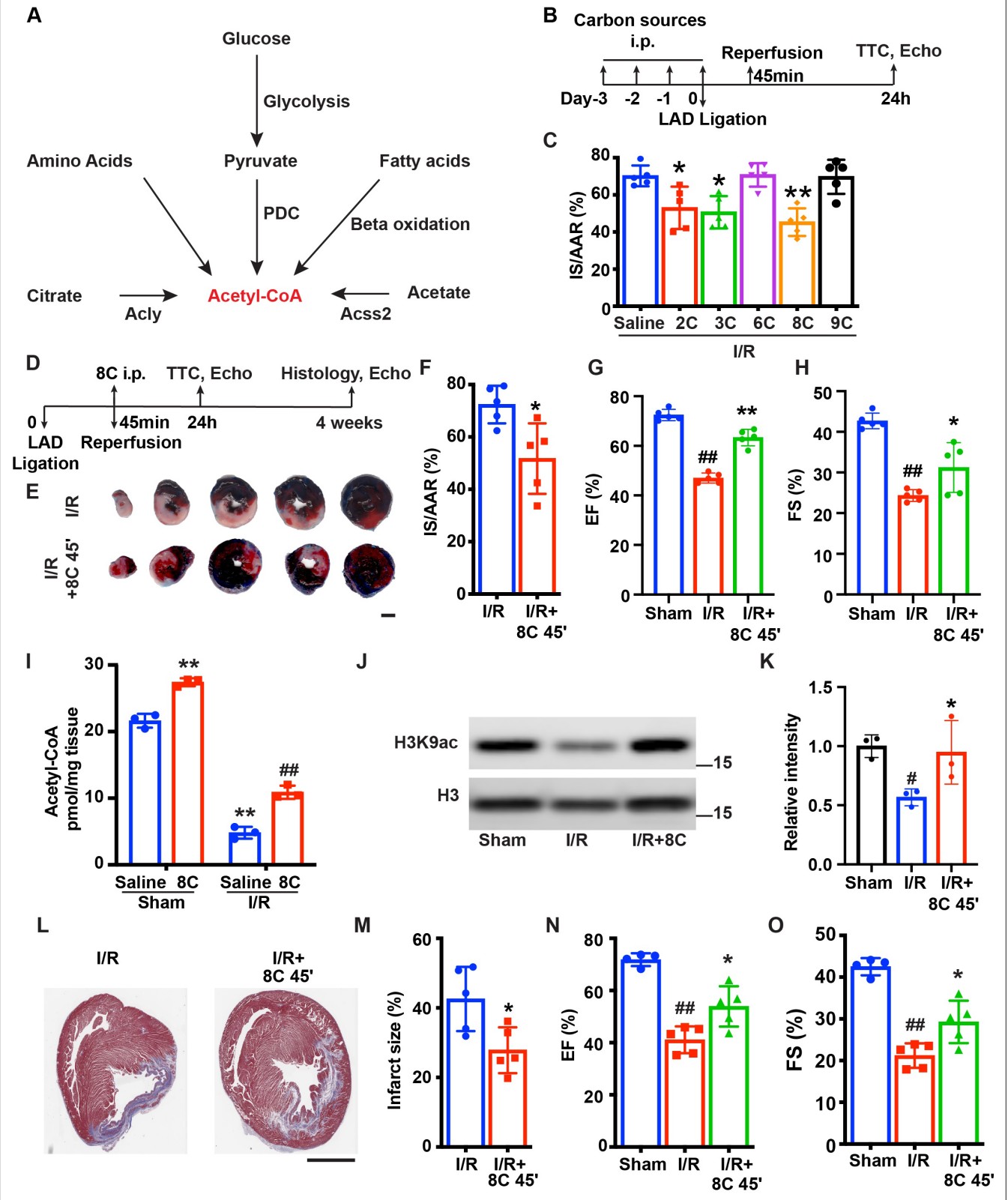

**Figure 1.** 8 C administration attenuates ischemia-reperfusion injury in rats. (**A**) Metabolic pathways of acetyl-CoA synthesis. (**B**) Schematic diagram of different carbon sources administration prior to I/R surgery. (**C**) Quantification of IS/AAR ratio in **Figure 1—figure supplement 1A** by Image J. (**D**) Schematic diagram of 8C administration post ischemic injury. (**E**) Representative figures of heart sections at 24 hr after I/R with or without 8C administration at reperfusion after 45 min ischemia. Scale bar: 2.5 mm (**F**) Quantification of IS/AAR ratio in (**E**). LV EF (**G**) and FS (**H**) at 24 hr after I/R.

*Figure 1 continued on next page*

Figure 1 continued

(I) Quantification of Acetyl-CoA levels in sham and I/R rat hearts at indicated conditions. (J–K) Western blot and quantification of H3K9ac and H3 in rat hearts 24 hr after I/R. (L) Trichrome masson staining of heart section after 4 weeks of I/R. Scale bar: 2.5 mm. (M) Quantification of infarct size in (I). LV EF (N) and FS (O) at 4 weeks after I/R. Error bars represent S.D. n = 5, #p < 0.05, ##p < 0.01 vs Sham group. * p < 0.05, **p < 0.01, vs I/R group. Data were analyzed by one-way ANOVA (C,G,H,K,N and O) or two-way ANOVA (I), followed by post-hoc Tukey test, data in F and M were analyzed by two-tailed student test.

The online version of this article includes the following source data and figure supplement(s) for figure 1:

Source data 1. Original numeric data for *Figure 1*.

Source data 2. Original western blot figure for *Figure 1*.

Figure supplement 1. Effect of metabolites on myocardial I/R injury.

was 45 min after LAD ligation. The infarct size was then measured by Evans blue/TTC staining at 24 hr after I/R (*Figure 1D*). Evans blue/TTC staining showed that 24 hr after reperfusion, 8C administration reduced the IS/AAR ratio from 72% to 52% compared to saline control after I/R (*Figure 1E–F* and *Figure 1—figure supplement 1C*). Moreover, 8C also significantly improved the left ventricle function evaluated by echocardiography 24 hr after I/R. Compared to saline treated I/R rats, 8C administration led to increase of EF and FS from 47% to 58% and 24% to 31%, respectively (*Figure 1G–H* and *Figure 1—figure supplement 1D*), while the normal rat heart has EF at 72% and FS at 43%. These results indicated that 8C significantly improved cardiac function by approximately 40%. Importantly, we found that there was a significant reduction of acetyl-CoA concentration in hearts after I/R, while 8C administration increased acetyl-CoA level in hearts after I/R (*Figure 1I*). Moreover, 8C administration elevated H3K9ac level in hearts after I/R (*Figure 1J–K*). Furthermore, to investigate whether a single dose of 8C administration is beneficial for long-term cardiac function after I/R, we examined the infarct size and heart function at 4 weeks after I/R. Trichrome Masson staining showed that the infarct size was notably reduced after 8C treatment (*Figure 1L, M*). The left ventricle function was also improved after 8C treatment as evidenced by the increase of EF and FS (*Figure 1N, O* and *Figure 1—figure supplement 1E*). Together with the fact that 8C can quickly enter into cells and contribute to around 50% of acetyl-CoA in heart at one hour after administration *Walton et al., 2003*, our results indicated that 8C administration at the time of reperfusion elevated acetyl-CoA production and histone acetylation level and significantly improved both short-term and long-term cardiac function after I/R.

## 8C attenuated cardiomyocyte apoptosis through alleviating oxidative stress

Apoptosis is one of the major reasons for cardiac damage after I/R injury *MacLellan and Schneider, 1997*. To detect the impact of 8C on cardiomyocyte apoptosis, TUNEL staining was performed at the border zone of infarct hearts. 8C treatment dramatically reduced TUNEL-positive cardiomyocytes in border zone (*Figure 2A, B*). Consistent with the TUNEL assay, the levels of myocardial death marker cTnI, serum CK-MB as well as total CK and LDH were also reduced in I/R rats after 8C administration (*Figure 2C, D* and *Figure 2—figure supplement 1A-B*). Moreover, 8C led to reduction in pro-apoptotic regulator Bax and upregulation of anti-apoptotic gene Bcl2 at 24 hr after I/R injury (*Figure 2E, F*). To study the mechanism of the beneficial effect of 8C administration after I/R, we examined the differential gene expression in the cells at the border zone 24 hr after I/R by RNA-seq (*Supplementary file 1*). Gene set enrichment analysis (GSEA) *Subramanian et al., 2005* showed that genes involved in apoptotic signaling pathways were enriched in saline-treated compared to 8C-treated rats after I/R (*Figure 2G*). Gene ontology (GO) analysis using differential expressed genes in 8C and saline-treated rats showed that genes related to cellular response to tumor necrosis factor were highly enriched in saline-treated I/R groups, whereas genes related to cardiomyocyte functions were highly expressed in 8C-treated I/R hearts (*Figure 2—figure supplement 1C-D*). These results suggested that 8C administration reduced the cardiomyocyte death after I/R. Specifically, I/R reduced the expression of anti-oxidative stress enzymes such as SOD1, SOD2, SOD3, CAT, Txnrd2, and Txnrd3, while 8C treatment upregulated the expression of these genes (*Figure 2—figure supplement 1E*). Since I/R-induced oxidative stress triggers CM apoptosis after reperfusion *Hausenloy and Yellon, 2013*, it is likely that 8C reduces cell death by activating antioxidant process after I/R injury.

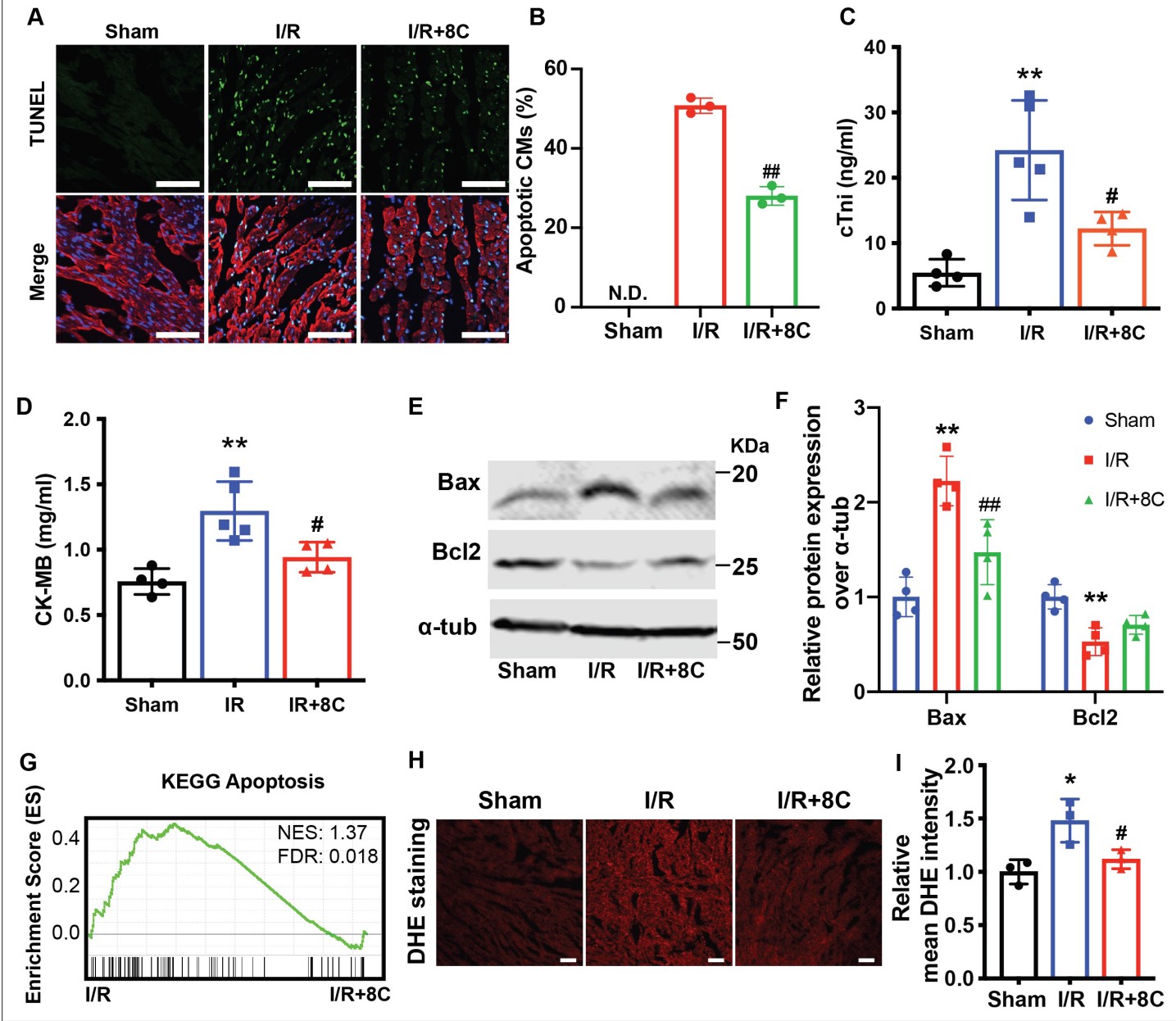

**Figure 2.** Post ischemic administration of 8C reduces the oxidative stress and cell death after I/R injury. (**A**) Representative of TUNEL (green) and cTnT (red) double staining at boarder zone at 24 hr post I/R injury. Scale bar: 100 µm (**B**) Quantification of cardiomyocytes cell death of 12 sections (**C–D**) Serum cTni and CK-MB level at 24 hr post I/R. (**E–F**) Western blot and quantification of Bax and Bcl2 at 24 hr after I/R. (**G**) GSEA analysis of Kegg apoptotic pathways after I/R with and without 8C treatment. (**H**) ROS levels were measured by DHE staining. Scale bar: 200 µm. (**I**) Relative mean DHE fluorescence intensity measured by Image J. Error bars represent S.D. n = 3. *p < 0.05, **p < 0.01 vs Sham; #p < 0.05, ##p < 0.01 vs I/R group. Data were analyzed by one-way ANOVA, followed by post-hoc Tukey test.

The online version of this article includes the following source data and figure supplement(s) for figure 2:

**Source data 1.** Original numeric data for *Figure 2*.

**Source data 2.** Original western blot figure for *Figure 2*.

**Figure supplement 1.** 8C administration altered the gene expression after I/R injury.

To address this, we measured the level of cardiac reactive oxygen species (ROS) after I/R by staining of dihydroethidium (DHE), a chemical that could be oxidized by ROS *Griendling et al., 2016*. The intensity of DHE signal was significantly lower in the presence of 8C after I/R (*Figure 2H, I*), indicating that 8C reduced oxidative stress after I/R. Moreover, 8C rescued the myocardial SOD activity after

I/R (*Figure 2—figure supplement 1F*). Altogether, these results showed that one major mechanism through which 8C improved cardiac function after I/R was reduction of the oxidative stress and subsequent cell apoptosis.

To further investigate the effect of 8C on oxidative stress and apoptosis in cardiomyocytes, neonatal rat ventricle myocytes (NRVMs) were subjected to simulated ischemia reperfusion (sI/R) with or without 8C. 8C or PBS was administered to NRVM culture medium for 12 hr. Then the NRVMs were subjected to 2 hr of simulated ischemia followed by 4 hr of simulated reperfusion. The percentage of apoptotic cells after sI/R was measured by labeling with Annexin V and Propidium iodide (PI). 8C significantly reduced the percentage of apoptotic cells, as evidenced by the lower percentage of Annexin V and PI double positive cells (*Figure 3A, B*). In addition, combination of CCK8 and LDH release assays were used to measure the total cell death *Fukami et al., 2016*. 8C administration reduced cell death after sI/R as showed by increased cell viability and decreased LDH release into to culture medium (*Figure 3C, D*). Moreover, the expression of cleaved PARP, which is a hallmark of apoptosis, was reduced with 8C administration (*Figure 3E*). Altogether, these results demonstrated that 8C reduced the apoptosis of NRVMs exposed to sI/R. To assess whether 8C reduced the oxidative stress in cardiomyocytes after sI/R, the accumulation of ROS level was measured by the intensity of DHE staining. The ROS level was significantly decreased in NRVMs with 8C treatment after sI/R (*Figure 3F-H*), indicating a direct effect of 8C in alleviating oxidative stress after sI/R. Thus, our collective in vivo and in vitro results revealed that 8C reduced cardiomyocyte apoptosis through alleviating oxidative stress after sI/R.

## Increasing acetyl-CoA synthesis by 8C administration stimulated histone acetylation and promoted antioxidative activity after I/R injury

8C-generated acetyl-CoA contributes to histone acetylation in several cell types *McDonnell et al., 2016*, and histone acetylation plays an important role in regulating cellular response to oxidative stress *Shimazu et al., 2013*. To examine whether 8C protected cardiomyocytes against I/R injury through stimulating histone acetylation, we first measured the acetyl-CoA level in NRVMs in vitro under sI/R. Similar to in vivo I/R injury, sI/R reduced acetyl-CoA level in NRVMs and 8C significantly increased the production of acetyl-CoA in NRVMs (*Figure 4A*). To determine the effect of acetyl-CoA replenishment on histone acetylation, we measured H3K9ac, H3K14ac, H3K27ac, and total H3 acetylation in NRVMs after sI/R. We found that sI/R led to a remarkable decrease of H3K9ac, H3K14ac, H3K27ac, and acH3, and that 8C increased histone acetylation in normal NRVMs and NRVMs under sI/R (*Figure 4B–E*). To further investigate whether the effects of different metabolites on I/R injury are related to histone acetylation, we examined the H3K9ac and acH3 levels in cardiomyocytes under sI/R in the presence of 2C, 3C, 6C, 8C, and 9C. We found that 2C, 3C, and 8C could lead to upregulation of acH3 and H3K9ac level which was consistent with our in vivo data that 2C, 3C and 8C but not 6C and 9C protected heart after I/R (*Figure 4—figure supplement 1B*). These results suggest that 8C, as well as 2C and 3C, likely reduced I/R injury through stimulating histone acetylation by acetyl-CoA production.

To further examine if 8 C derived acetyl-CoA directly contribute to histone acetylation, we applied C13 tracing experiments. Myocytes were cultured with [U-13C] sodium octanoate for 12 hr, and the histone proteins were purified for post-translational modification analysis using LC-MS/MS. A significant amount of acetyl histones showed mass shifts in MS spectra, indicating that 8 C derived acetyl-CoA contributed to histone acetylation (*Figure 4F*). To determine if 8 C could directly contribute to histone acetylation in animal heart, we injected mice with 160 mg/kg [U-13C] sodium octanoate before I/R surgery and isolated heart tissue 30 min after reperfusion for acetylation analysis. Indeed, C13 labeled acetyl-histones were detected in in vivo heart after MI (*Figure 4—figure supplement 1D*).

Acetyl-CoA is the substrate for histone acetyltransferases (HATs) to generate histone acetylation by transferring the acetyl-group from acetyl-CoA to histone lysine residues *Henry et al., 2015*. Specifically, HATs with low affinity to acetyl-CoA are more sensitive to acetyl-CoA abundance. H3K9ac has been reported as the histone acetylation most sensitive to acetyl-CoA levels *Henry et al., 2015*. Consistent with this finding, we found that 8 C led to most significant changes in H3K9ac after sI/R. Thus, we reasoned that H3K9ac, which is enriched in promoters for gene activation, is one key epigenetic event for gene regulation after sI/R. To examine the potential epigenetic regulation of 8 C derived acetyl-CoA in antioxidative stress, we performed ChIP to measure H3K9ac at the promoters of antioxidant genes. 8 C elevated H3K9ac level at the promoters of NQO1, HO1, and SOD2 in both

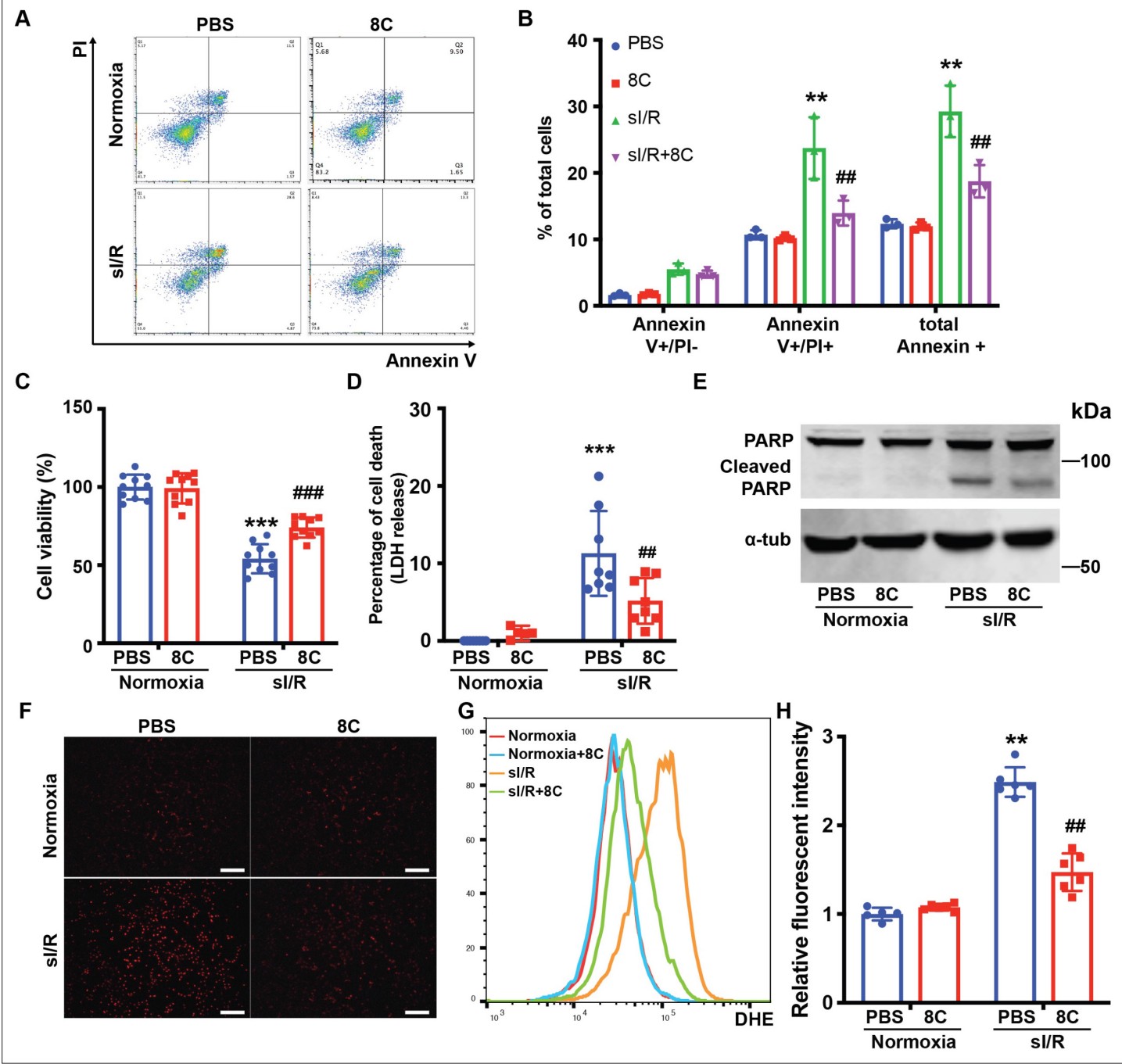

**Figure 3.** 8C attenuates NRVMs apoptosis through reducing oxidative stress. (**A**) FACS analysis of Annexin V and PI staining in NRVMs exposed to sI/R with and without 8C treatment. (**B**) Quantification of percentage of Annexin V + and PI+ cells. Cell viability and cell death measurement in NRVMs with sI/R using CCK8 detection kit (**C**) and LDH assay kit (**D**). (**E**) Western blot of cleaved PARP in NRVMs after sI/R treatment. (**F**) NRVM cellular ROS levels are indicated by DHE staining after sI/R treatment. Scale bar: 200 µm (**G**) FACS analysis of NRVM DHE staining after sI/R. (**H**) Relative mean fluorescence intensity of DHE staining measured by Flowjo. n = 3, **p < 0.01, ***p < 0.001, vs Normoxia + PBS; ## p < 0.01, ###p < 0.001 vs sI/*R* + PBS. Data were analyzed by two-way ANOVA, followed by post-hoc Tukey test.

The online version of this article includes the following source data for figure 3:

**Source data 1.** Original numeric data for *Figure 3*.

**Source data 2.** Original western blot figure for *Figure 3*.

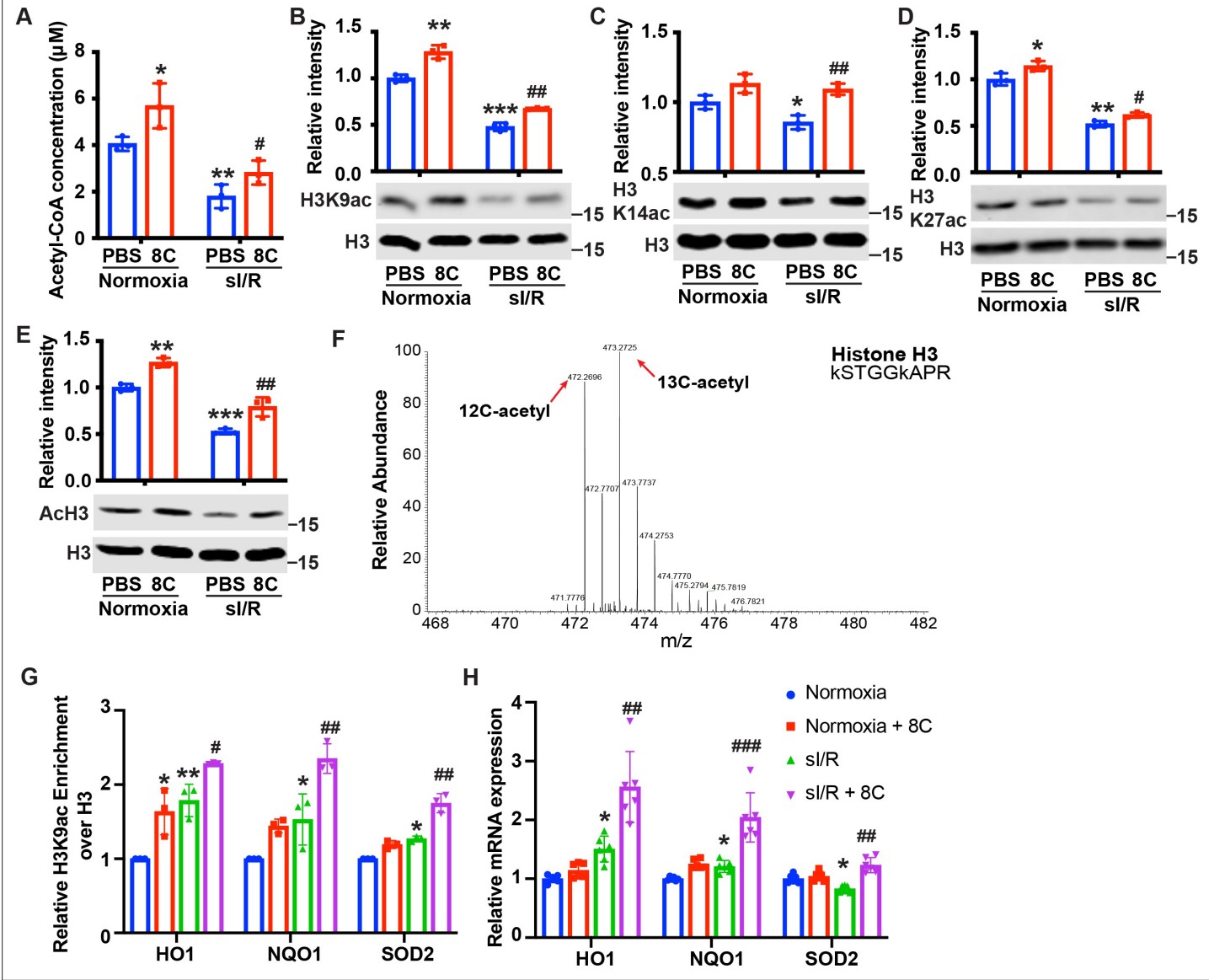

**Figure 4.** 8C stimulates histone acetylation and promotes antioxidant gene expression. (**A**) Quantification of Acetyl-CoA concentrations in NRVMs subjected to sI/R. (**B–E**) 8 C rescues sI/R reduced H3K9ac, H3K27ac, H3K14ac and total acH3 levels. NRVMs were treated with or without 0.5 mM 8 C under sI/R. The histone acetylation levels were determined by western blot. Total H3 in the same blot was used as loading control. (**F**) Representative MS spectra from NRVMs treated with [U-13C] sodium octanoate indicating isotope distribution on acetyl Histone H3. The mass shifts indicate the incorporation of heavy carbon to the acetyl histone lysines. (**G**) Enrichment of H3K9ac over H3 at promoters of HO1, NQO1, and SOD2 in NRVMs after sI/R. (**H**) mRNA expression of HO1, NQO1, and SOD2 in NRVMs after sI/R. *p < 0.05, **p < 0.01, ***p < 0.001, vs Normoxia + PBS; #p < 0.05, ##p < 0.01, ###p < 0.001 vs sI/*R* + PBS. Data were analyzed by two-way ANOVA, followed by post-hoc Tukey test.

The online version of this article includes the following source data and figure supplement(s) for figure 4:

**Source data 1.** Original numeric data for *Figure 4*.

**Source data 2.** Original western blot figure for *Figure 4*.

**Figure supplement 1—source data 1.** Original western blot figure for *Figure 4—figure supplement 1*.

**Figure supplement 1.** 8C administration altered the gene expression after I/R injury.

normoxia and sI/R conditions, with a more profound increase under sI/R (*Figure 4G*). 8 C upregulated both mRNA and protein expression of antioxidant genes including HO1, NQO1, and SOD2 after sI/R (*Figure 4H* and *Figure 4—figure supplement 1F*), indicating that the histone acetylation increase by 8 C promoted antioxidant response under stress. Collectively, our results suggested that 8C-produced acetyl-CoA contributed to epigenetic regulation of antioxidant genes in response to I/R injury.

## MCAD was required for the conversion of 8C into acetyl-CoA and subsequent histone acetylation increase and heart protection

To ascertain whether 8 C produced acetyl-CoA contributed tothe rescue of histone acetylation after sI/R, we knocked down MCAD (*Figure 5—figure supplement 1A*), a key enzyme in the generation of acetyl-CoA from 8C *Matsubara et al., 1986*. Knockdown of MCAD disrupted the metabolism of 8 C, and therefore led to reduction of histone acetylation promoted by 8 C in NRVMs in both normoxia and sI/R condition (*Figure 5A*). Thus, these data indicated that metabolic production of acetyl-CoA from 8 C was required for the 8C-mediated histone acetylation regulation. To determine whether acetyl-CoA mediated histone acetylation was key to the 8 C heart protective effect after I/R, we examined the cardiomyocyte survival after MCAD knockdown with and without 8 C after sI/R. MCAD knockdown significantly blocked the protective effect of 8 C after sI/R, as evidenced by decreased cell viability and increased LDH release in MCAD knockdown cells in presence of 8 C after sI/R compared to knockdown controls with 8 C treatment after sI/R (*Figure 5B* and *Figure 5—figure supplement 1B*). Importantly, MCAD knockdown blocked the 8C-reduced cellular ROS level after sI/R (*Figure 5C–D* and *Figure 5—figure supplement 1C*). Specifically, we found that MCAD knockdown reduced 8 C stimulated H3K9ac increase in the promoters of HO1 and NQO1 after sI/R (*Figure 5E*). Subsequently, MCAD knockdown reduced 8C-elevated expression of HO1 and NQO1 after sI/R (*Figure 5F* and *Figure 5—figure supplement 1D-F*). In addition, the treatment of 8 C did not affect the efficiency of MCAD knockdown. Thus, these results demonstrated that MCAD-mediated 8 C metabolism was essential for histone acetylation and attenuating apoptosis through activating antioxidative process.

## HAT enzyme Kat2a was required for 8C-mediated histone acetylation to inhibit oxidative stress in heart protection

HATs are the very enzymes that catalyze histone acetylation by transferring the acetyl-group from acetyl-CoA to histone lysine residues. As H3K9ac is most sensitive to physiological acetyl-CoA levels, we then hypothesized that HATs that acetylate H3K9 are the most responsive to physiological acetyl-CoA concentrations would play important roles in the cardioprotection of 8 C. Kat2a and Kat2b are the major HATs that modulate H3K9ac *Jin et al., 2011*. Kat2a is mostly responsive to acetyl-CoA concentrations at 0–10 μM *Wang et al., 2017*, while Kat2b is response to acetyl-CoA in the 0–300 μM range *Shi et al., 2016*. Kat2a expression was upregulated in rat heart after I/R injury, whereas 8 C treatment diminished Kat2a upregulation (*Figure 6—figure supplement 1A*). Together with the fact that acetyl-CoA levels in NRVMs ranged from 1 to 7 μM under sI/R (*Figure 4A*), we focused on studying the effect of Kat2a knockdown (*Figure 6—figure supplement 1B*) on the protective role of 8 C after sI/R. Kat2a knockdown largely abolished the H3K9ac increase caused by 8 C treatment in NRVMs (*Figure 6A*), indicating that Kat2a was a key HAT to mediate 8C-stimulated histone acetylation. Furthermore, Kat2a knockdown led to a significant decrease of cell viability and increase of LDH release compared to knockdown control group in presence of 8 C after sI/R (*Figure 6B* and *Figure 6—figure supplement 1C*), suggesting that Kat2a was required for the protective effect of 8 C. Moreover, knockdown of Kat2a abolished 8 C's effect on inhibiting the cellular ROS level after sI/R (*Figure 6C–D* and *Figure 6—figure supplement 1D*). Specifically, Kat2a knockdown abolished 8C-stimulated H3K9ac increase at the promoters of HO1 and NQO1 after sI/R (*Figure 6E*). Subsequently, Kat2a knockdown reduced 8C-elevated expression of HO1 and NQO1 after sI/R (*Figure 6F* and *Figure 6—figure supplement 1E-G*). These results illustrated that Kat2a was required to execute the rescuing role of 8 C after sI/R by modulating histone acetylation, which in turn activated antioxidant gene expression and attenuated cellular apoptosis after sI/R. Together our investigation revealed an integrated metabolic and epigenetic network comprising 8 C, acetyl-CoA, MCAD, and Kat2a, that likely played an essential role in combating heart injury after I/R (*Figure 6G*).

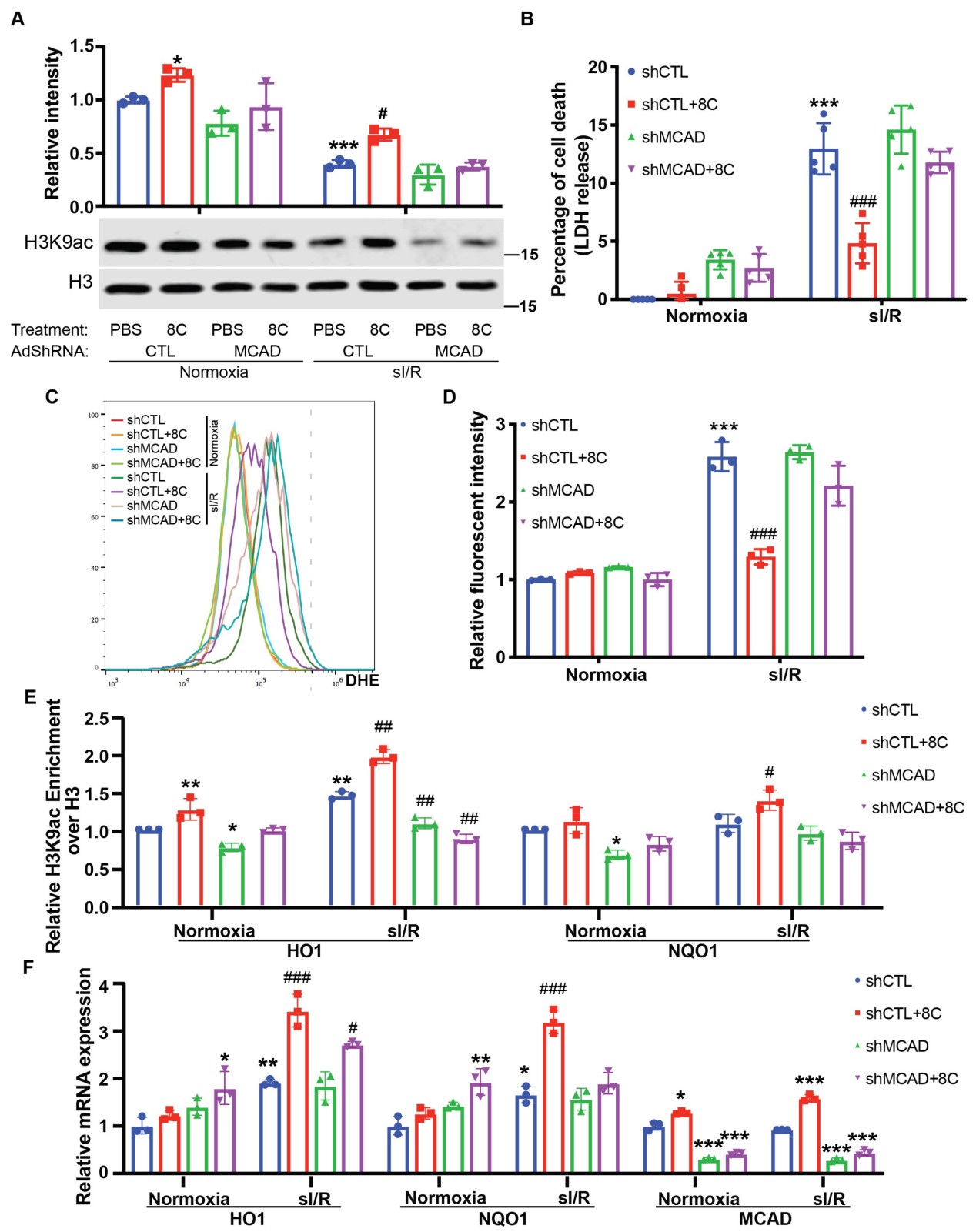

**Figure 5.** MCAD was required for the conversion of 8 C into acetyl-CoA and subsequent histone acetylation increase and heart protection. (**A**) Western blot of H3K9ac level showed MCAD knockdown reduced 8C-induced H3K9ac increase in NRVMs under both normoxia and sI/R. (**B**) Measurement of medium LDH levels in NRVMs at indicated condition using LDH assay kit. (**C**) FACS analysis of DHE staining NRVMs after sI/R. (**D**) Relative mean fluorescence intensity of DHE staining. (**E**) Enrichment of H3K9ac over H3 at promoters of HO1 and NQO1 after sI/R at indicated conditions. (**F**) mRNA

*Figure 5 continued on next page*

*Figure 5 continued*

expression of HO1, NQO1 and MCAD in NRVMs after sI/R. n = 3, *p < 0.05, **p < 0.01, ***p < 0.001, vs Normoxia + PBS + shCTL; #p < 0.05, ##p < 0.01, ###p < 0.001 vs sI/*R* + PBS + shCTL. Data were analyzed by two-way ANOVA, followed by post-hoc Tukey test.

The online version of this article includes the following source data and figure supplement(s) for figure 5:

**Source data 1.** Original numeric data for *Figure 5*.

**Source data 2.** Original western blot figure for *Figure 5*.

**Figure supplement 1.** Knockdown of MCAD is required for alleviating ROS accumulation.

**Figure supplement 1—source data 1.** Original western blot figure for *Figure 5—figure supplement 1*.

## Discussion

In this study, we have established an interlinked metabolic and epigenetic network comprising 8 C, acetyl-CoA, MCAD, and Kat2a that stimulates histone acetylation and anti-oxidative stress gene expression to combat heart injury. Our screen of acetyl-CoA synthesis metabolites identifies that 2 C, 3C, and 8 C administration significantly improves cardiac function after I/R injury. Specifically, we establish that induction of acetyl-CoA synthesis by 8 C metabolism stimulates histone acetylation and promotes cardiomyocyte survival after I/R. Our study further reveals that 8C-stimulated histone acetylation leads to increase of antioxidant gene expression for heart repair. Moreover, MCAD knockdown diminishes the 8C-induced acetylation and subsequently lowers antioxidant activity after sI/R in vitro, indicating that the metabolic conversion of 8 C to acetyl-CoA is mainly responsible for histone acetylation and subsequent heart repair effect. Furthermore, the effect of 8 C on heart repair through acetyl-CoA and subsequent nuclear histone acetylation is evidenced by Kat2a studies, as Kat2a knockdown largely diminishes the protective effect of 8 C after sI/R. Our study demonstrates systematically for the first time that modulating acetyl-CoA abundance can determine cardiomyocyte response to I/R injury via common metabolic and epigenetic mechanisms.

Our study reveals a novel mechanism centered on acetyl-CoA that connects metabolic dynamics and epigenetic regulation in cardiac repair after I/R injury and suggests that acetyl-CoA could be a survival signal for cardiomyocyte after I/R injury. Our data show that I/R injury reduces the cellular acetyl-CoA level, which is associated with decrease of histone acetylation and increase of cardiomyocyte death after I/R injury *Granger et al., 2008*. We further show that these associations are causally related. In particular, we demonstrate that 8 C restores the acetyl-CoA level and subsequently increases the histone acetylation and improves cardiac function after injury. Moreover, 8 C administration directly contributes about 50% of acetyl-CoA in rat hearts *Walton et al., 2003*, implying that 8 C derived acetyl-CoA plays an essential role after I/R injury. Importantly, knockdown of MCAD, which metabolizes 8 C to acetyl-CoA, diminishes the rescuing effect of 8 C treatment. Moreover, other acetyl-CoA sources including acetate and pyruvate that improved cardiac function after I/R could also stimulate histone acetylation in cardiomyocytes under sI/R in vitro. Thus, our study suggests that acetyl-CoA from 8 C metabolism, as well as from acetate and pyruvate, could improve heart function after I/R through modulating histone acetylation.

Our study indicates that histone acetylation is a major downstream event of 8 C and acetyl-CoA in heart repair after injury, which is consistent with the notion that acetyl-CoA could serve as second messenger to modulate epigenetic response to environmental changes *Pietrocola et al., 2015*. Indeed, tracing by C13-labeled 8 C indicate that 8 C produced acetyl-CoA directly contributes to histone acetylation. Knockdown of Kat2a, a major HAT enzyme in catalyzing histone acetylation, greatly diminishes the cardiomyocyte protective effect of 8 C metabolism under a sI/R setting. Consistent with this notion, 8 C effect on elevating histone acetylation and heart protection is largely abolished under Kat2a knockdown conditions. Although 8C-mediated acetyl-CoA synthesis could promote mitochondrial metabolism for cardiac protection, our Kat2a knockdown experiments suggest that 8C-mediated histone acetylation in gene expression plays a key role in cardiac protection. Histone acetylation as a general epigenetic regulatory mechanism can play essential roles in numerous cellular processes *Backs and Olson, 2006*. In this study, we have identified that 8C-mediated histone acetylation increases the expression of HO1, NQO1 and SOD2 and decreases the ROS level after I/R injury both in vivo and in vitro. These data indicate that the protecting effect of 8 C treatment after heart injury is at least partially mediated through stimulating gene expression against oxidative stress. This result is consistent with the previous observation that a high level of histone acetylation activates gene

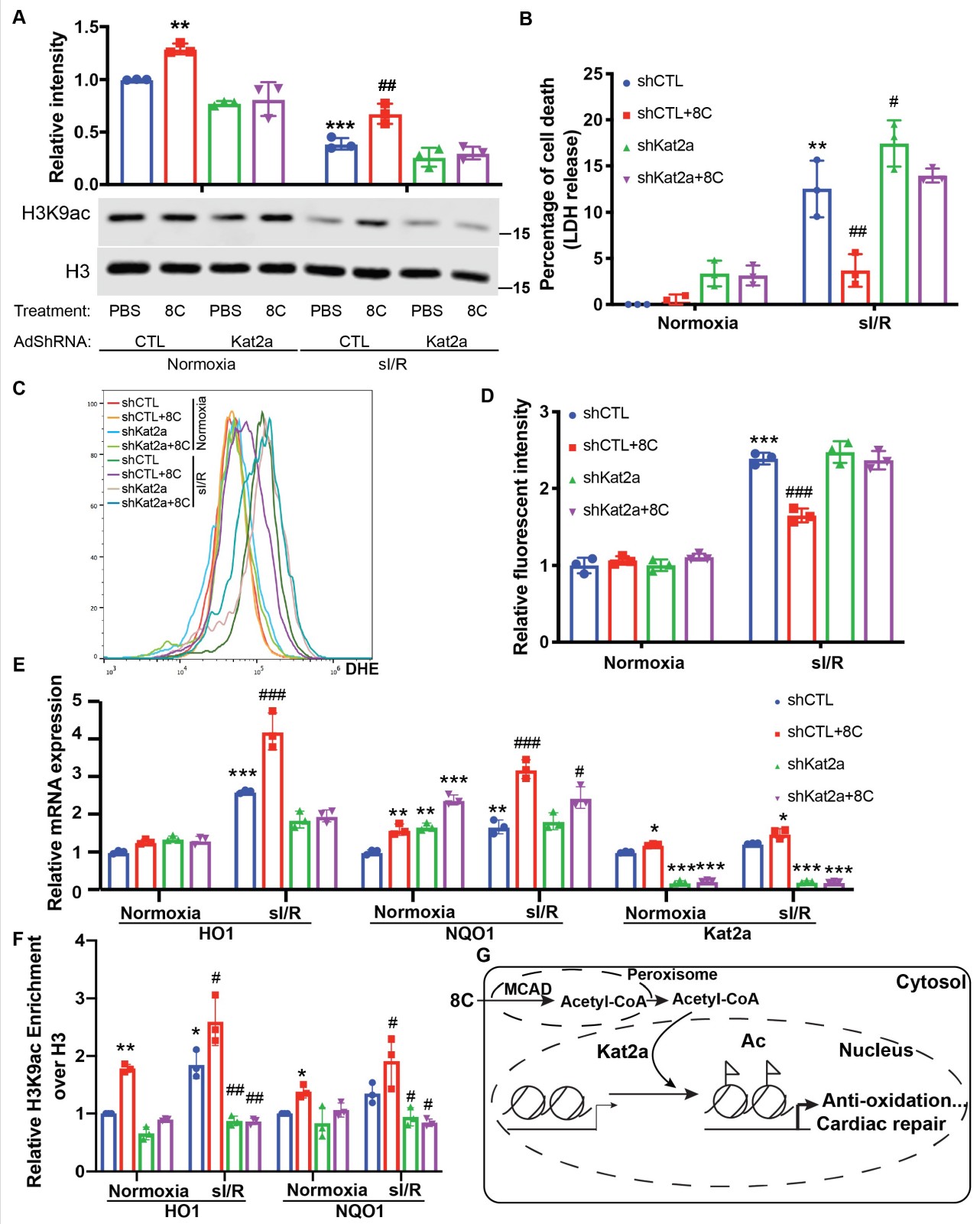

**Figure 6.** HAT enzyme Kat2a was required for 8 C mediated histone acetylation to inhibit oxidative stress in heart protection. (**A**) Western blot of H3K9ac level showed Kat2a knockdown reduced 8C-induced H3K9ac increase in NRVMs under both normoxia and sI/R. (**B**) Measurement of medium LDH levels in NRVMs at indicated condition using LDH assay kit. (**C**) FACS analysis of DHE staining NRVMs after sI/R. (**D**) Relative mean fluorescence intensity of DHE staining. (**E**) Enrichment of H3K9ac over H3 at promoters of HO1 and NQO1 after sI/R at indicated conditions. (**F**) mRNA expression

*Figure 6 continued on next page*

*Figure 6 continued*

of HO1,NQO1 and Kat2a in NRVMs after sI/R. (**G**) Schematic diagram of 8 C metabolism for cardiac repair. n = 3, *p < 0.05, **p < 0.01, ***p < 0.001, vs Normoxia + PBS + shCTL; #p < 0.05, ##p < 0.01, ###p < 0.001 vs sI/*R* + PBS + shCTL. Data were analyzed by two-way ANOVA, followed by post-hoc Tukey test.

The online version of this article includes the following source data and figure supplement(s) for figure 6:

**Source data 1.** Original numeric data for *Figure 6*.

**Source data 2.** Original western blot figure for *Figure 6*.

**Figure supplement 1.** Knockdown of Kat2a is required for alleviating ROS accumulation.

**Figure supplement 1—source data 1.** Original western blot figure for *Figure 6—figure supplement 1*.

expression against oxidative stress *Shimazu et al., 2013*. A future genome-wide study will provide a complete picture of 8C-mediated cellular processes in combating heart injury.

Histone acetylation can also be regulated by histone deacetylase (HDACs) *Backs and Olson, 2006*. Interestingly, studies by us and others reveal that chemical inhibition of HDACs also leads to attenuation of myocardial infarction *Tian et al., 2019*; *Xie et al., 2014* and heart failure *Jeong et al., 2018*. Our recent work indicates that valproic acid, an FDA approved HDAC inhibitor for treatment bipolar disorder, protects heart function after I/R injury by promoting a Foxm1-mediated transcriptional pathway *Tian et al., 2019*. In addition, SAHA, another HDAC inhibitor, blunts myocardial infarction via regulating autophagy activities *Xie et al., 2014*. Moreover, HDAC inhibitors could alter the acetylation of myofibrils and govern diastolic function of heart *Jeong et al., 2018*. Whether acetyl-CoA and HDAC inhibitor mediated histone acetylation alternations share the same regulatory mechanisms in cardiac repair requires detailed investigation. It will also be interesting to determine how metabolism mediated histone acetylation and HDAC inhibition coordinate their actions in cardiac repair.

Our establishment of direct connection between epigenetic status and metabolite abundance in heart repair may provide an alternative perspective for numerous published studies. For instance, activation of AMPK shows cardioprotective effect after I/R injury *Russell et al., 2004*. It is postulated that the ability of AMPK in enhancing glucose uptake *Russell et al., 2004* and suppressing ribosome biogenesis *Cao et al., 2017* is the major reason for cardioprotection. However, the translation of AMPK activation for cardiac therapy has not been successful, partially due to limited understanding of the mechanism for AMPK protection. A recent study shows that activation of AMPK results in increased level of acetyl-CoA and therefore likely elevates histone acetylation *Salminen et al., 2016*. Thus, it is possible that AMPK promotes cardiac repair via direct epigenetic regulation. Similarly, altering the levels of different metabolites show beneficial effect on cardiac repair *Haar et al., 2014*; *Olenchock et al., 2016*. The change of Acetyl-CoA may also have a direct effect on the expression of HATs after I/R injury in heart. Kat2a, a major HAT enzyme in catalyzing histone acetylation, was upregulated after I/R, while enrichment of acetyl-CoA by 8 C metabolism diminished the need of Kat2a upregulation. Considering the potential roles of these metabolites in epigenetic regulations, re-examining these studies will likely provide new insights into metabolite-mediated epigenetic changes in heart disease treatment beyond the mitochondria function, and lead to more successful therapy.

Our study suggests that increasing histone acetylation by metabolic acetyl-CoA production is an effective strategy for heart repair. Although increasing histone acetylation by metabolic acetyl-CoA production is an effective strategy for heart repair, not all metabolites that generate acetyl-CoA and histone acetylation have the same beneficial effect. One major reason could be that certain metabolites also generate other signals that cancel or even outweigh this beneficial effect. For example, succinate is found to be the major source of ROS production after I/R and succinate administration aggravates I/R injury *Chouchani et al., 2014*; *Zhang et al., 2018*. 9 C produces succinate through anaplerotic reaction, and our results show that the accumulation of succinate level is much higher in 9 C compared to 8 C treatment (*Figure 1—figure supplement 1F*). Moreover, 9 C also produces propionyl-CoA, which inhibits the activity of Kat2a *Montgomery et al., 2015*, and likely abolishes the potential effect of acetyl-CoA on stimulating histone acetylation. Together, these data may explain the null effect of 9 C administration in heart repair. Overall, this study elucidates that exploring detailed metabolic and epigenetic mechanisms mediated by various metabolic carbon sources in combating I/R injury will be an exciting research area to develop potential effective heart therapies.

# Materials and methods

**Key resources table**

| Reagent type (species) or resource | Designation | Source or reference | Identifiers | Additional information |
|---|---|---|---|---|
| Antibody | Bcl2 (Rabbit monoclonal antibody) | Abcam | Cat#: ab182858; RRID: AB_2715467 | Western blot (1:1000) |
| Antibody | Bax (Rabbit polyclonal antibody) | Cell Signaling Technology | Cat#: 2,772 S; RRID: AB_10695870 | Western blot (1:1000) |
| Antibody | α-Tubulin (mouse monoclonal antibody) | Cell Signaling Technology | Cat#: 3,873 S; RRID: AB_1904178 | Western blot (1:1000) |
| Antibody | PARP (Rabbit polyclonal antibody) | Cell Signaling Technology | Cat#: 9542; RRID: AB_2160739 | Western blot (1:1000) |
| Antibody | H3K9ac (Rabbit polyclonal antibody) | Sigma | Cat#: 06–942; RRID: AB_310308 | Western blot (1:3000), 1:100 for ChIP (1:100) |
| Antibody | H3K14ac (Rabbit polyclonal antibody) | Sigma | Cat#: 07–353; RRID: AB_310545 | Western blot (1:3000) |
| Antibody | H3K27ac (Rabbit polyclonal antibody) | Millipore | Cat#: 07–360; RRID: AB_310550 | Western blot (1:3000) |
| Antibody | AcH3 (Rabbit polyclonal antibody) | Millipore | Cat#: 06–599; RRID: AB_2115283 | Western blot (1:3000) |
| Antibody | Histone H3 (mouse monoclonal antibody) | Sigma | Cat#: 14,269 S; RRID: AB_2756816 | Western blot (1:2000) |
| Antibody | HO1 (Rabbit polyclonal antibody) | Proteintech Group Inc | Cat#: 10701–1-AP; RRID: AB_2118685 | Western blot (1:1000) |
| Antibody | NQO1 (Rabbit polyclonal antibody) | Proteintech Group Inc | Cat#: 11451–1-AP; RRID: AB_2298729 | Western blot (1:1000) |
| Antibody | SOD2 (Rabbit polyclonal antibody) | Proteintech Group Inc | Cat#: 24127–1-AP; RRID:AB_2879437 | Western blot (1:1000) |
| Antibody | MCAD (Rabbit polyclonal antibody) | Proteintech Group Inc | Cat#: 55210–1-AP; RRID: AB_10837361 | Western blot (1:1000) |
| Antibody | Kat2a (Rabbit recombinant antibody) | Abcam | Cat#: Ab217876; RRID:AB_2811191 | Western blot (1:1000) |
| Antibody | cTnT (Rabbit polyclonal antibody) | Abcam | Cat#: ab45932; RRID: AB_956386 | IF (1:200), FACS (1:200) |
| Antibody | 680RD Donkey anti-Mouse IgG (H + L) | LI-COR | Cat# 926–68072; RRID:AB_10953628 | Western blot (1:5000) |
| Antibody | 800CW Donkey anti-Rabbit IgG (H + L) | LI-COR | Cat# 926–32213; RRID:AB_621848 | Western blot (1:5000) |
| Chemical compound, drug | Sodium acetate | Sigma | Cat#:S2889-250G | |
| Chemical compound, drug | Sodium Pyruvate | Sigma | Cat#:P2265-5G | |
| Chemical compound, drug | Sodium Citrate | Sigma | Cat#: PHR1416-1G | |
| Chemical compound, drug | Sodium Octanoate | Fisher | Cat#: N029125G | |
| Chemical compound, drug | Sodium Nonanoate | TCI America | Cat#: N0291-25G | |
| Chemical compound, drug | Evans Blue | Fisher | Cat#: AC195550250 | |

*Continued on next page*

*Continued*

| Reagent type (species) or resource | Designation | Source or reference | Identifiers | Additional information |
|---|---|---|---|---|
| Chemical compound, drug | 2,3,5-Triphenyltetrazolium chloride | Sigma | Cat#: T8877 | |
| Chemical compound, drug | Dihydroethidium (DHE) | Cayman | Cat#: 12,013 | |
| Chemical compound, drug | U-13C-Sodium octanoate | Cambridge Isotope Laboratories | Cat#:CLM-9617-PK | |
| Chemical compound, drug | EDTA-free Protease Inhibitor Cocktail | Sigma | Cat#:118350001 | |
| Chemical compound, drug | Type II Collagenase | Worthington Biochemical Co. | Cat#: LS004174 | |
| Chemical compound, drug | 2-deoxy-D-Glucose | Cayman | Cat#: 14,325 | |
| Peptide, recombinant protein | Micrococcal Nuclease | NEB | Cat#: M0247S | |
| Chemical compound, drug | IHC Zinc Fixative | Fisher | Cat#: BDB550523 | |
| Commercial assay or kit | Creatine Kinase Activity Assay Kit | Sigma | Cat#: MAK116-1KT | |
| Commercial assay or kit | Lactate Dehydrogenase Activity Assay Kit | Sigma | Cat#: MAK066 | |
| Commercial assay or kit | SOD Assay Kit-WST | Dojindo | Cat#: S311-10 | |
| Commercial assay or kit | PowerUp SYBR Green Master Mix | Thermo Fisher | Cat#: A25778 | |
| Commercial assay or kit | Precision Plus Protein All Blue Prestained | Bio-rad | Cat#:1610373 | |
| Commercial assay or kit | iScript cDNA Synthesis Kit | Bio-rad | Cat#:1708891 | |
| Commercial assay or kit | AccuPrep PCR Purification Kit | Bioneer | Cat#: K-3034 | |
| Commercial assay or kit | Trichrome Stain (Masson) Kit | Sigma | Cat#: HT15-1KT | |
| Commercial assay or kit | TUNEL Assay Kit | Sigma | Cat#:11684795910 | |
| Commercial assay or kit | Cytotoxicity LDH Assay Kit-WST | Dojindo | Cat#: CK12-05 | |
| Commercial assay or kit | PicoProbeAcetyl-CoA Fluorometric Assay Kit | Fisher | Cat#: NC9976028 | |
| Commercial assay or kit | Cell Counting Kit-8 | Dojindo | Cat#: CK04-13 | |
| Commercial assay or kit | BD Annexin V-FITC | Fisher | Cat#: BDB556420 | |
| Commercial assay or kit | Succinate Colorimetric Assay Kit | Fisher | Cat#: NC0541966 | |
| Commercial assay or kit | TRIzol Reagent | Sigma | Cat#: 15596018 | |

*Continued*

| Reagent type (species) or resource | Designation | Source or reference | Identifiers | Additional information |
|---|---|---|---|---|
| Commercial assay or kit | TURBO DNase | Thermo Fisher | Cat#: AM2238 | |
| Commercial assay or kit | NEBNext Ultra II Directional RNA Library Prep Kit for Illumina | NEB | Cat#: E7760L | |
| Commercial assay or kit | Creatine kinase (CK) MB isolenzyme Elisa kit | ABclonal | RK03571 | |
| Commercial assay or kit | Troponin I (TnI) Elisa kit | Abclonal | RK03995 | |
| Strain, strain background (*Escherichia coli*) | NEB stable E-coli | NEB | Cat#: C3040H | |
| Strain, strain background (*Escherichia coli*) | NEB 10-beta competent *E. coli* | NEB | Cat#:C3019H | |
| Recombinant DNA reagent | BLOCK-iT U6 RNAi Entry Vector Kit | Invitrogen | Cat#: K494500 | |
| Recombinant DNA reagent | Gateway LR Clonase II Enzyme Mix | Thermo Fisher | Cat#: 11791–020 | |
| Recombinant DNA reagent | pAd/PL-DEST Gateway Vector Kit | Thermo Fisher | Cat#: V49420 | |
| Cell line (Homo-sapiens) | 293 A Cell Line | Thermo Fisher | R70507 | For package of Adenovirus |
| Software, algorithm | ImageJ | NIH | https://imagej.nih.gov/ij/ | |
| Software, algorithm | GraphPad Prism (version 7) | Graphpad Software | https://www.graphpad.com/scientific-software/prism/ | |
| Software, algorithm | Odyssey CLx Imaging System | LI-COR Biosciences | https://www.licor.com/ | |
| Software, algorithm | FlowJo | Flowjo | https://www.flowjo.com/ | |

## Animal experiments

All experiments were approved by the Institutional Animal Care and Use Committee of the University of Michigan (PRO00009606) and were performed in accordance with the recommendations of the American Association for the Accreditation of Laboratory Animal Care.

## Generation of rat I/R models

Myocardial ischemia/reperfusion was carried out in rats as described previously *Tian et al., 2019*. Briefly, 8–9 weeks old male SD rats (Charles River Laboratories) were anesthetized with ketamine (100 mg/kg) and xylazine (10 mg/kg). Myocardial ischemia was performed by occlusion of the left anterior descending coronary artery (LAD) using 6–0 silk sutures. After 45 min of ischemia, the myocardium was reperfused. Sodium acetate (2 C), sodium pyruvate (3 C), sodium citrate(6 C), sodium octanoate (8 C), and sodium nonanoate (9 C) were dissolved in saline at concentrations of 125 mg/mL, 125 mg/mL, 125 mg/mL, 40 mg/mL, and 50 mg/mL, respectively. 2 C (500 mg/kg) (*Frost et al., 2014*), 3 C (500 mg/kg) (*González-Falcón et al., 2003*), 6 C (500 mg/kg), 8 C (160 mg/kg) (*Vinay et al., 1976*), and 9 C (200 mg/kg) (*Vinay et al., 1976*) were intraperitoneal (i.p.) injected to rats for 3 continuous days and another dose before LAD ligation for screening assay. In a clinically relevant setting, 8 C was i.p. injected at the time of reperfusion, which was 45 min after LAD ligation. For blood sample

collection, the animals were anesthetized with ketamine (100 mg/kg) and xylazine (10 mg/kg) then euthanized by injection of pentobarbital at 50 mg/kg for subsequent heart isolation.

## Echocardiography

The rats were anesthetized with 2% isoflurane. Echocardiography (ECG) was performed 1 day and 4 weeks after surgery using Vevo 2,100 system with M model. Left ventricular internal diameter end diastole (LVIDd) and left ventricular internal diameter end systole (LVIDs) were measured perpendicularly to the long axis of the ventricle. Ejection fraction (EF) and fractional shortening (FS) were calculated according to LVIDd and LVIDs.

## Evans blue/triphenyltetrazolium chloride (TTC) staining

After 24 hr reperfusion, the LAD was re-occluded and 5% Evans blue was injected into the right ventricle. The heart was then removed, frozen rapidly and sliced into five 2 mm transverse sections. The sections were incubated with 1% TTC in phosphate buffer (pH 7.4) at 37 °C for 10 min and photographed by EPSON scanner. The ischemia area (IS), area at risk (AAR), and left ventricular area, were measured with Image J software.

## Measurement of serum CK, serum LDH, and tissue SOD

Blood samples were collected at 24 hr after reperfusion and plasma was isolated. The level of serum troponin I (TnI) and activity of creatine kinase (CK) MB isolenzyme, total CK and lactate dehydrogenase (LDH) in plasma were measured using commercial kits (TnI: Abclonal,RK03995; CK-MB:Abclonal, RK03571; CK:Sigma, MAK116-1KT; LDH: Sigma, MAK066-1KT) according to the manufacturer's instructions. Ventricles were crushed to a powder using liquid nitrogen and homogenized in saline with the weight/volume ratio of 1:10. After centrifuging for 10 min at 3500 rpm, the supernatants were withdrawn for SOD activity measurement using SOD assay kit-WST (Dojindo) according to the manufacturer's instructions. Bradford protein assay was performed to determine the protein concentration.

## Histology assay

Histological studies were performed as previously described *Li et al., 2017*. Briefly, animals were sacrificed, and the hearts were perfused with 20% KCl. After being fixed with zinc fixative solution (BD Pharmingen) and dehydrated by alcohol, the samples were embedded by paraffin and sectioned into 5 μm slides. The sections were processed for immunostaining, including Masson's trichrome, immunofluorescence and TUNEL assay (in situ cell death detection kit, Roche). Images were captured by Aperio (Leica Biosystems, Buffalo Grove, IL, USA) and a confocal microscope (Nikon, Melville, NY, USA) and analyzed by Image J software.

## Cell lines

The 293 A cells were pre-authenticated and purchased from Thermo scientific.

NRVMs were isolated from postnatal day 1–3 SD rats as previously described *Gao et al., 2018*. Briefly, neonatal rat hearts were minced into small pieces and transferred into conical tube. The tissues were digested in Trypsin-EDTA (0.25%) at 4 °C overnight, then subjected to 1 mg/mL type II collagenase at 37 °C for 30 min. The cell suspension was collected and then centrifuged at 1000 rpm for 5 min. The resultant pellet was resuspended, and the neonatal cardiac fibroblasts were removed by 2 repeats of 45 min plating in the cell incubator. The enriched NRVMs were then cultured in 5% horse serum for 2 days then changed into serum-free medium before further analysis. The purity of NRVMs were assessed by FACS using cTnT antibody.

All cell cultures were examined for mycoplasma every two weeks to make sure no mycoplasma contamination.

## Simulated ischemia reperfusion(sI/R) of NRVMs in vitro

NRVMs were cultured in simulated ischemia medium *Xie et al., 2014* and subjected to hypoxia in a chamber with 94% $N_2$,1% $O_2$, 5% $CO_2$. After 2 hr of hypoxia, the cells were then reperfused in DMEM for 4 hr in serum free medium at 95% air and 5% $CO_2$.

## Measurement of acetyl-CoA

Acetyl-CoA was measured using acetyl-CoA assay kit (Biovision) according to the manufacturer's instructions. For tissues, hearts were weighted and pulverized, then mixed with 400 μL of 1 M perchloric acid per 100 mg tissue. For cell culture, 5 million NRVMs were isolated and lysed in RIPA buffer. The lysate was deproteinized by 1 M perchloric acid. The deproteinized supernatant was neutralized by 3 M KHCO$_3$. The supernatant was then measured acetyl-CoA following the standard kit protocol. The concentration of acetyl-CoA in cells was calculated with the estimation of average NRVM size as 6000 μm$^3$ *Yang et al., 2018*; *Bensley et al., 2016*.

## Western blot

Proteins were extracted in RIPA lysis buffer (NaCl 150 mM, NP-40 1%, Sodium deoxycholate 0.5%, SDS 0.1%, Tris 25 mM) followed by centrifugation at 4 °C for 15 min at 12,000 rpm. Protein concentration was measured by Bradford protein assay and 40 μg of total protein was separated by SDS-PAGE and then transferred to PVDF membranes. The membranes were blocked with 5% nonfat dry milk for 1 hr at room temperature and then incubated with primary antibodies overnight at 4 °C. After three washings with TBST, the membranes were incubated with secondary antibody in TBST solution for 1 hr at room temperature. After three washings, the membranes were scanned and quantified by Odyssey CLx Imaging System (LI-COR Biosciences, USA).

## RNA-Seq

Total RNA was extracted from the LV tissues using Trizol following manufacture's protocol. The total RNA was purified with DNAse I to remove genomic DNA. RNA quality was assessed using Agilent Bioanalyzer Nano RNA Chip. One μg of total RNA (RIN > 8) was used to prepare the sequencing library using NEBNext Stranded RNA Kit with mRNA selection module. The library was sequenced on illumina HiSeq 4,000 (single end, 50 base pair) at the Sequencing Core of University of Michigan. The RNA-seq data have been deposited in Gene Expression Omnibus with the accession code GSE132515.

## RNA-Seq data analysis

RNA-seq data was quantified using Kallisto (Version 0.43.0) *Bray et al., 2016* with parameters: `--single` -b 100 l 200 s 20 using the Rnor6.0 (ensembl v91). The estimated transcript counts were exported by tximport *Soneson et al., 2015* for Deseq2 analysis *Love et al., 2014*. Differential expression was then calculated using Deseq2 default setting. Gene Ontology analysis was performed using GSEA *Subramanian et al., 2005*.

## DHE staining

For in vitro staining, 5 mM of freshly prepared DHE solution was directly added to medium to final concentration of 5 μM and cultured at 37 °C for 30 min. The cells were washed three times of PBS and observed under microscope or dissociated for FACS analysis.

For in vivo staining, heart tissues were embedded in OCT immediately after harvested. Tissues were then sectioned at 20 μm. 5 μM of DHE solution was directly applied to sections for 30 min at 37 °C. After three washes of PBS, the sections were mounted and observed under the microscope.

## ChIP

ChIP experiments were performed as previously described *Lei et al., 2015*. Cells were fixed with 1% formaldehyde for 10 min and quenched with 0.125 M glycine. Nuclei pellets were then harvested and digested with micrococcal nuclease at 37 °C for 2 min. After brief sonication, chromatin solution was incubated with Dynabeads and antibodies against H3K9ac or pan H3 control overnight. Beads were washed four times with LiCl wash buffer (0.25 M LiCl, 1% IGEPAL CA630, 1% sodium deoxychulate, 1 mM EDTA, 10 mM Tris, pH 8.1) and one wash of TE buffer then eluted with elution buffer at 65°C. DNA was purified using Bioneer PCR purification kit. Enrichment of immunoprecipitated DNA was normalized to pan H3 by quantitative PCR.

## Mass spectrometry analysis of C13-labeled histone acetylation

For in vitro analysis, NRVMs were treated with C13-labeled sodium octanoate (0.5 mM) for 12 hr before histone purification. For in vivo study, the C13-labeled sodium octanoate was injected at

dose of 160 mg/kg at time of perfusion, the histone proteins were purified using LV tissue 30 min after reperfusion. The purified histones were digested by Arg-C proteinase and followed by a 90-min LC-MS/MS data acquisition for posttranslational modification analysis *Anwar et al., 2018*.

## Statistical analysis

GraphPad Prism Software (version 7) was used for statistical analysis. Data were expressed as the mean ± SD. Statistical comparisons between two groups were performed by Student's t test, and more than two groups were performed by two-way or one-way ANOVA followed by post-hoc Tukey comparison. Groups were considered significantly different at $p < 0.05$.

## Acknowledgements

This work was supported by National Institutes of Health (NIH) of United States (HL109054 and HL139735), an Inaugural Grant from the Frankel Cardiovascular Center, an MCube Grant from University of Michigan, and a Pilot Grant from the University of Michigan Health System – Peking University Health Sciences Center Joint Institute for Clinical and Translational Research to ZW. We thank Frankel Cardiovascular Center Physiology and Phenotyping Core for performing all echocardiography examination. We thank the Proteomics Resource Facility in University of Michigan for performing the Mass Spectrometry analysis.

## Additional information

### Funding

| Funder | Grant reference number | Author |
|---|---|---|
| National Institutes of Health | HL109054 | Zhong Wang |
| National Institutes of Health | HL139735 | Zhong Wang |
| University of Michigan | Intramural Grant | Zhong Wang |

The funders had no role in study design, data collection and interpretation, or the decision to submit the work for publication.

### Author contributions

Ienglam Lei, Conceptualization, Data curation, Formal analysis, Investigation, Writing – original draft, Writing – review and editing; Shuo Tian, Wenbin Gao, Data curation, Investigation, Writing – review and editing; Liu Liu, Yijing Guo, Investigation, Writing – review and editing; Paul Tang, Eugene Chen, Writing – review and editing; Zhong Wang, Conceptualization, Funding acquisition, Supervision, Writing – review and editing

### Author ORCIDs

Ienglam Lei (iD) http://orcid.org/0000-0002-0248-6871
Wenbin Gao (iD) http://orcid.org/0000-0001-9268-8823
Zhong Wang (iD) http://orcid.org/0000-0002-8720-4609

### Ethics

All experiments were approved by the Institutional Animal Care and Use Committee of the University of Michigan (PRO00009606) and were performed in accordance with the recommendations of the American Association for the Accreditation of Laboratory Animal Care.

### Decision letter and Author response

Decision letter https://doi.org/10.7554/eLife.60311.sa1
Author response https://doi.org/10.7554/eLife.60311.sa2

## Additional files

### Supplementary files
- Supplementary file 1. List of DEGs between I/R and I/R with 8 C treatment.
- Transparent reporting form

### Data availability
The RNA-seq data have been deposited in Gene Expression Omnibus with the accession code GSE132515.

The following dataset was generated:

| Author(s) | Year | Dataset title | Dataset URL | Database and Identifier |
|---|---|---|---|---|
| Wang Z | 2021 | Acetyl-CoA production by specific metabolites promotes cardiac repair after myocardial infarction via histone acetylation | http://www.ncbi.nlm.nih.gov/geo/query/acc.cgi?acc=GSE132515 | NCBI Gene Expression Omnibus, GSE132515 |

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
