## [Editor Report]

The authors present evidence for a novel role of acetyl CoA in response to different carbon sources and histone acetylation in the context of myocardial ischemia-reperfusion injury following myocardial infarction. The findings will add to the understanding of mechanisms underlying the pathophysiology of reperfusion injury and introduce novel potential targets for therapeutic intervention.

---

## [Decision Letter]

**Decision letter after peer review:**

Thank you for submitting your article "Acetyl-CoA production by select metabolites promotes cardiac repair after myocardial infarction via histone acetylation" for consideration by *eLife*. Your article has been reviewed by 3 peer reviewers, one of whom is a member of our Board of Reviewing Editors, and the evaluation has been overseen by a Reviewing Editor and a Senior Editor. The reviewers have opted to remain anonymous.

The reviewers have discussed the reviews with one another and the Reviewing Editor has drafted this decision to help you prepare a revised submission.

Summary:

The manuscript by Lei and colleagues explores the role of acetyl-CoA in mediating the metabolic/epigenetic crosstalk that is proposed to occur in heart injury. Specifically, they focus on defining the source of the acetyl-CoA and which specific histone modifications it contributes to. Using both the rat LAD-surgery induced myocardial I/R model and an in vitro NRVM model of ischemic injury they propose that sodium octanoate (8C) is the source and that it leads to transcriptional reprogramming of antioxidant and apoptosis genes through histone acetylation mediated by lysine acetyltransferase 2a (KAT2A). Importantly, they also show that the source of this 8C is via MCAD and that this can contribute to heart repair. Overall, this is an important area of research. However, the reviewers have several concerns regarding this manuscript.

Essential revisions:

1. The main hypothesis of this study is that increasing mitochondrial acetyl CoA production will increase histone acetylation which may protect the heart from I/R injury. However, despite the fact that acetylation of proteins in each compartment is strongly linked to local acetyl-CoA levels (PMID: 30467427), the authors did not take into consideration the cell compartment-specific level of acetyl-CoA in their study. In this case, while the nuclear/cytoplasm acetyl CoA levels could be critical for histone acetylation, the mitochondrial acetyl CoA levels may not have a direct impact on histone acetylation but rather on mitochondrial protein acetylation. This is a major methodological gap and the authors should include specifically the changes in nuclear/cytoplasmic acetyl CoA levels.

2. It is generally accepted that increased acetylation of certain histone proteins improves cardiovascular function in response to various injuries which led to the discovery of histone deacetylase (HDAC) inhibitors. In contrary, hyperacetylation of several mitochondrial proteins are linked with metabolic disturbances and cardiac dysfunction (PMID: 32413386). Relevant to the current study, there is strong evidence suggesting that increased acetylation of antioxidant enzymes in the mitochondria including SOD2 is associated with its decreased activity and mitochondrial dysfunction. However, this concept was not considered despite the fact that increased acetyl CoA production may have a negative impact on these enzymes by hyperacetylating them. How do the authors reconcile this contradiction? In addition, how do they explain the absent of mitochondrial acetylation impact assessment in their study?

3. Most of the carbon sources used in this study were not the common ones or physiologically relevant to human FA metabolism. Why did the authors exclude physiologically relevant fatty acids such as palmitate and oleate in their study?

4. Despite all of the metabolites contributing to acetyl CoA generation, the authors tried to show that only the acetyl CoA derived from 8C is responsible for increased histone acetylation. However, the subsequent experiments confirm this 8C specific acetyl CoA action is problematic. Firstly, fractional contribution of each metabolite to the total acetyl CoA pool was not accessed. Though sophisticated, carbon labeling and metabolite analysis from tissue extracts by NMR could be used to accurately determine this. But the KD of MCAD can also block metabolism of other metabolites catalyzed by this enzyme and is not specific to 8C. Thus, there is no direct evidence to support that acetyl CoA from 8C is specifically responsible for increased histone acetylation. At least the authors could have compared the effect of each of these metabolites on the histone acetylation level instead of 8C only.

5. As indicated in #2 above, increased mitochondrial acetyl CoA may not be the exact mechanism for increased histone acetylation following these short chain FA administration. Rather, as suggested previously, direct binding of the short chain fatty acids to HDACS and its inhibition could be possible mechanisms that could have been investigated. Any of nuclear cytoplasmic specific acetyl CoA generating enzymes: ATP-citrate lyase (ACLY), and acyl-coenzyme A synthetase short-chain family member 2 (ACSS2), and the nuclear pyruvate dehydrogenase complex (PDC), could have been also targeted instead of MCAD to better understand the situation.

6. There are evidence that shows that fatty acid oxidation is increased during reperfusion, which implies an increased acetyl CoA level (PMID: 9293951). However, in this study the authors suggested that there is a decrease in acetyl CoA level after I/R. How do the authors reconcile this paradox?

7. The use of serum CK and LDH as an indicator of cell death is also misleading. First, these are not the best markers. Secondly, instead of measuring cardiac specific isoenzymes, the authors analyzed only unspecified(total?) serum levels of these enzymes.

8. To understand the pathophysiological significance of the findings the authors should measure the concentrations of the metabolites that are achieved after IP injection.

9. Conclusions would be strengthened if heavy labeled substrates were used in the in vivo experiments and the authors showing that the acetyl-coA that they are measuring in heart tissues were derived from label that was injected.

10. Although 8C was the most statistically significant protection, Figure 1C, the changes were similar with 2C and 3C. This should be expanded upon in the discussion as other sources of acetyl-CoA could also be protective.

11. Please include a supplementary table with the genes and their fold change for the pathway analysis; e.g. Figure 2G. Along those lines there is a comment in the Figure 2 legend for a heatmap (H) of the antioxidant gene expression that is not provided. In place of it is the DHE staining images

12. The data in Figure 4 is robust. However, please reword the conclusion ending on page 10. Some of the decreases in histone acetylation was rescued, however not all (e.g. H3K9 and H3K27) which were attenuated. Please rephrase.

13 There is a disconnect between histone acetylation, RNA, and protein for the targets examined in Figure 4 and S3; i.e. HO1 and NQO1. Although this is not uncommon, it does draw into question the proposed transcriptional regulation mechanism versus post-transcriptional regulation. Specifically, how is the increased sI/R acetylation in these targets associated with increased mRNA but decreased protein and the further increase in histone acetylation seen with sI/R+8C is leading to a further increase in mRNA but rescue of protein, while the increase in acetylation in normoxia+8C has no effect. Please clarify or adjust conclusions based on the data shown.

14. Inclusion of Mcad in Figure 5F is an important control to show that 8C is not interfering with the KD. Please discuss in the text.

15. A primary conclusion of the paper is the role of KAT2A. The in vitro work in Figure 6 supports this. However, the authors should also present if Kat2a gene expression was changed in the RNA-seq data from the rat model. It is noted that KAT2A protein could be driving the differences in vivo based on acetyl-CoA levels but this should be discussed.

16. Some of western blot band are shown in different molecular weight than it should be normally. First, cleaved caspase 3 band in figure 3E is shown to be under 20kda while the band in supplementary figure S6 is shown around 25kda. As cleaved caspase 3 is normally detected around 17 or 19 KDa, this cleaved caspase 3 blot may be nonspecific band from the antibody used in western blot. Second, NQO1 band in figure S3, S4, S5 and S8 are shown to be above 37 KDa while normally NQO1 is detected around 29-31 KDa. This NQO1 band may be non-specific band from the antibody. The authors are advised to confirm the western blot results for cleaved caspase 3 and NQO1.

[Editors' note: further revisions were suggested prior to acceptance, as described below.]

Thank you for resubmitting your work entitled "Acetyl-CoA production by select metabolites promotes cardiac repair after myocardial infarction via histone acetylation" for further consideration by *eLife*. Your revised article has been evaluated by a Reviewing Editor in consultation with the original reviewers and a Senior Editor.

The manuscript has been improved but there are some remaining issues that need to be addressed, as outlined below:

1) In Suppl. Figure 6A, authors should describe. that IR + 8C reduced Kat2a expression. Seems paradoxical. This should also be discussed.

2) Echocardiography – Describe the device/machine used and indicate the anesthesia that was used if any.

3) What is the basis for estimated NRVM surface area as 6000μm3? This should be measured directly, or validation of this approach shown.

4) Numerous grammatical errors exist, and the manuscript should be editorially revised by an English language editor.

---

## [Author Response]

Essential revisions:1. The main hypothesis of this study is that increasing mitochondrial acetyl CoA production will increase histone acetylation which may protect the heart from I/R injury. However, despite the fact that acetylation of proteins in each compartment is strongly linked to local acetyl-CoA levels (PMID: 30467427), the authors did not take into consideration the cell compartment-specific level of acetyl-CoA in their study. In this case, while the nuclear/cytoplasm acetyl CoA levels could be critical for histone acetylation, the mitochondrial acetyl CoA levels may not have a direct impact on histone acetylation but rather on mitochondrial protein acetylation. This is a major methodological gap and the authors should include specifically the changes in nuclear/cytoplasmic acetyl CoA levels.

Thanks for raising this question. Our study is based on previous research indicating that octanoate (8C) can be metabolized by MCAD in peroxisome to generate cytoplasmic/nuclear acetyl-CoA^1^. To directly assess the contribution of 8C derived acetyl-CoA to histone acetylation, we performed C13 labeled octanoate tracing experiments both in vitro and in vivo. We found a direct contribution of acetyl-CoA (13C2) to acetyl-histone in both cases. The new results are presented in Figure 4F and Figure 4—figure supplement 1C-D. The results provide direct evidence showing that 8C derived acetyl-CoAs are used as substrates for histone acetylation and greatly strengthen our conclusion.

2. It is generally accepted that increased acetylation of certain histone proteins improves cardiovascular function in response to various injuries which led to the discovery of histone deacetylase (HDAC) inhibitors. In contrary, hyperacetylation of several mitochondrial proteins are linked with metabolic disturbances and cardiac dysfunction (PMID: 32413386). Relevant to the current study, there is strong evidence suggesting that increased acetylation of antioxidant enzymes in the mitochondria including SOD2 is associated with its decreased activity and mitochondrial dysfunction. However, this concept was not considered despite the fact that increased acetyl CoA production may have a negative impact on these enzymes by hyperacetylating them. How do the authors reconcile this contradiction? In addition, how do they explain the absent of mitochondrial acetylation impact assessment in their study?

We thank the reviewer for this comment. The antioxidant enzymes are regulated by both transcriptionally and post-translationally, and as the reviewer suggests, the reduced activities of antioxidant enzymes by potential hyperacetylation may cancel the effect of transcriptional upregulation of antioxidant enzymes by histone acetylation increase. The overall effect of acetyl-CoA production on antioxidant activity would be a sum of the two effects. In our experiments, we found the treatment of 8C could reduce oxidation level in the cells. Thus, it is likely that the effect of 8C on histone acetylation is greater than potential acetylation of antioxidant enzymes and overall shows a beneficial effect. This is also the main reason we did not examine the changes in mitochondrial acetylation in our study.

3. Most of the carbon sources used in this study were not the common ones or physiologically relevant to human FA metabolism. Why did the authors exclude physiologically relevant fatty acids such as palmitate and oleate in their study?

The palmitate and oleate require CD36 for transport. The CD36 expression was reduced after I/R injury. Therefore, we use medium chain fatty acids that can permeabilize to cells to bypass the involvement of CD36 for membrane transfer. Nevertheless, we will also examine these long chain fatty acids in our future studies.

4. Despite all of the metabolites contributing to acetyl CoA generation, the authors tried to show that only the acetyl CoA derived from 8C is responsible for increased histone acetylation. However, the subsequent experiments confirm this 8C specific acetyl CoA action is problematic. Firstly, fractional contribution of each metabolite to the total acetyl CoA pool was not accessed. Though sophisticated, carbon labeling and metabolite analysis from tissue extracts by NMR could be used to accurately determine this. But the KD of MCAD can also block metabolism of other metabolites catalyzed by this enzyme and is not specific to 8C. Thus, there is no direct evidence to support that acetyl CoA from 8C is specifically responsible for increased histone acetylation. At least the authors could have compared the effect of each of these metabolites on the histone acetylation level instead of 8C only.

We thank the reviewer for the suggestion. We have performed experiments on the effect of other metabolites on histone acetylation, the revised figures are now presented in Figure S3A-B. Moreover, the newly C13 tracing experiments also provide direct evidence of the contribution of acetyl-CoA from 8C to histone acetylation (Figure 4F and Figure 4—figure supplement 1C-D).

5. As indicated in #2 above, increased mitochondrial acetyl CoA may not be the exact mechanism for increased histone acetylation following these short chain FA administration. Rather, as suggested previously, direct binding of the short chain fatty acids to HDACS and its inhibition could be possible mechanisms that could have been investigated. Any of nuclear cytoplasmic specific acetyl CoA generating enzymes: ATP-citrate lyase (ACLY), and acyl-coenzyme A synthetase short-chain family member 2 (ACSS2), and the nuclear pyruvate dehydrogenase complex (PDC), could have been also targeted instead of MCAD to better understand the situation.

Our C13 tracing experiment shows that 8C derived acetyl-CoA directly contributes to histone acetylation. Therefore, it is more reasonable to examine the metabolic enzymes of 8C that convert 8C to acetyl-CoA.

The 8C is known to be converted to Acetyl-CoA in peroxisome and directly contribute to nucleus-cytoplasmic acetyl-CoA pool. Therefore, we used MCAD knockdown to determine if the contribution of 8C to histone acetylation is through the metabolic process of 8C.

We did not include the study of Acly, Acss2, and PDC in this manuscript, because their substrates are not direct metabolic product of 8C. We plan to report the roles of these enzymes in contributing to acetyl-CoA for histone acetylation after myocardial I/R in future studies.

6. There are evidence that shows that fatty acid oxidation is increased during reperfusion, which implies an increased acetyl CoA level (PMID: 9293951). However, in this study the authors suggested that there is a decrease in acetyl CoA level after I/R. How do the authors reconcile this paradox?

We thank the reviewer for this comment. The paper reviewer referenced suggests the inhibition activity of ACC, an enzyme determines the ratio of acetyl-CoA: Malonyl-CoA, not directly the level of acetyl-CoA. The ratio could increase by either increased level of acetyl-CoA or decreased level of Malonyl-CoA. There is a study directly measuring the levels of acetyl-CoA and Malonyl-CoA, which shows that both acetyl-CoA and Malonyl-CoA levels decrease after I/R, whereas the ratio of acetyl-CoA vs Malonyl-CoA significantly increases ^2^.

7. The use of serum CK and LDH as an indicator of cell death is also misleading. First, these are not the best markers. Secondly, instead of measuring cardiac specific isoenzymes, the authors analyzed only unspecified(total?) serum levels of these enzymes.

We detect that total CK in the original studies. We have performed experiments to detect cTnI and CK-MB, the cardiac specific enzyme activity in the serum. The new results are added in revised Figure 2C-D and Figure 2—figure supplement 1A-B. We have provided the catalog number in the methods. In addition to these serum markers, we also perform TUNEL and western blot of Bax/Bcl2 in heart tissues to study the cell death differences among different treatments. Altogether, these results could provide a confident comparison of cell death.

8. To understand the pathophysiological significance of the findings the authors should measure the concentrations of the metabolites that are achieved after IP injection.

We thank the reviewer for the comments. The concentration of octanoate and its metabolites after injection has been well documented in previous studies^3, 4^. Therefore, we did not measure the concentration of metabolites, and use the concentration of 8C determined in the previous studies.

9. Conclusions would be strengthened if heavy labeled substrates were used in the in vivo experiments and the authors showing that the acetyl-coA that they are measuring in heart tissues were derived from label that was injected.

To address this excellent comment, we injected C13 label Octanoate after MI. We then performed the C13 tracing of histone acetylation. The contribution of C13 acetyl-CoA to histone was identified in the heart tissue Figure 4F and Figure 4—figure supplement 1C-D.

10. Although 8C was the most statistically significant protection, Figure 1C, the changes were similar with 2C and 3C. This should be expanded upon in the discussion as other sources of acetyl-CoA could also be protective.

Thanks, we expanded our discussion with the beneficial role of 2C and 3C after I/R.

11. Please include a supplementary table with the genes and their fold change for the pathway analysis; e.g. Figure 2G. Along those lines there is a comment in the Figure 2 legend for a heatmap (H) of the antioxidant gene expression that is not provided. In place of it is the DHE staining images

We apologize for the error. We provided the missing heatmap in the original manuscript in Figure 2—figure supplement 1E. We also provided the differential expression gene list with the fold change in newly provided excel table as supplementary file 1.

12. The data in Figure 4 is robust. However, please reword the conclusion ending on page 10. Some of the decreases in histone acetylation was rescued, however not all (e.g. H3K9 and H3K27) which were attenuated. Please rephrase.

We have rephrased the word to rescue. Thank you for this suggestion.

13 There is a disconnect between histone acetylation, RNA, and protein for the targets examined in Figure 4 and S3; i.e. HO1 and NQO1. Although this is not uncommon, it does draw into question the proposed transcriptional regulation mechanism versus post-transcriptional regulation. Specifically, how is the increased sI/R acetylation in these targets associated with increased mRNA but decreased protein and the further increase in histone acetylation seen with sI/R+8C is leading to a further increase in mRNA but rescue of protein, while the increase in acetylation in normoxia+8C has no effect. Please clarify or adjust conclusions based on the data shown.

We thank the reviewer for this comment. We have revised our results to clarify the histone change by 8C and expression upregulation under sI/R. The new description is in the last paragraph on page 10.

14. Inclusion of Mcad in Figure 5F is an important control to show that 8C is not interfering with the KD. Please discuss in the text.

Thanks, we have included Mcad knockdown in the text.

15. A primary conclusion of the paper is the role of KAT2A. The in vitro work in Figure 6 supports this. However, the authors should also present if Kat2a gene expression was changed in the RNA-seq data from the rat model. It is noted that KAT2A protein could be driving the differences in vivo based on acetyl-CoA levels but this should be discussed.

Thanks. Based on the RNA-seq results, Kat2a expression is upregulated about 1.6 folds after I/R. We have presented this result in revised Figure 5—figure supplement 1A.

16. Some of western blot band are shown in different molecular weight than it should be normally. First, cleaved caspase 3 band in figure 3E is shown to be under 20kda while the band in supplementary figure S6 is shown around 25kda. As cleaved caspase 3 is normally detected around 17 or 19 KDa, this cleaved caspase 3 blot may be nonspecific band from the antibody used in western blot. Second, NQO1 band in figure S3, S4, S5 and S8 are shown to be above 37 KDa while normally NQO1 is detected around 29-31 KDa. This NQO1 band may be non-specific band from the antibody. The authors are advised to confirm the western blot results for cleaved caspase 3 and NQO1.

We thank the reviewer for pointing out this issue. We did the cleaved caspase 3 WB and can only observe band above 20KD. To avoid potential nonspecific band and misleading results, we have removed the Cleaved Caspase 3 results from Figure 3. We analyzed the NQO1 western blot and saw there was a band with correct size at ~30 KD. We reported the NQO1 band at 30KD in Figure 4—figure supplement 1E, Figure 5—figure supplement 1D, Figure 6—figure supplement 1E. We also revised the quantification of blots according to band intensity with proper size.

References

1. Bian F, Kasumov T, Thomas KR, Jobbins KA, David F, Minkler PE, Hoppel CL and Brunengraber H. Peroxisomal and mitochondrial oxidation of fatty acids in the heart, assessed from the 13C labeling of malonyl-CoA and the acetyl moiety of citrate. *J Biol Chem*. 2005;280:9265-71.

2. Kudo N, Barr AJ, Barr RL, Desai S and Lopaschuk GD. High rates of fatty acid oxidation during reperfusion of ischemic hearts are associated with a decrease in malonyl-CoA levels due to an increase in 5'-AMP-activated protein kinase inhibition of acetyl-CoA carboxylase. *J Biol Chem*. 1995;270:17513-20.

3. Walton ME, Ebert D and Haller RG. Octanoate oxidation measured by 13C-NMR spectroscopy in rat skeletal muscle, heart, and liver. *J Appl Physiol (1985)*. 2003;95:1908-16.

4. Kimura I, Inoue D, Maeda T, Hara T, Ichimura A, Miyauchi S, Kobayashi M, Hirasawa A and Tsujimoto G. Short-chain fatty acids and ketones directly regulate sympathetic nervous system via G protein-coupled receptor 41 (GPR41). *Proc Natl Acad Sci U S A*. 2011;108:8030-5.

[Editors' note: further revisions were suggested prior to acceptance, as described below.]

The manuscript has been improved but there are some remaining issues that need to be addressed, as outlined below:1) In Suppl. Figure 6A, authors should describe. that IR + 8C reduced Kat2a expression. Seems paradoxical. This should also be discussed.

We revised the description in the Results section to state “Kat2a expression was upregulated in rat heart after I/R injury, whereas 8C treatment diminished Kat2a upregulation (Figure 6—figure supplement 1A).” We speculate that under I/R condition, Kat2a is upregulated in order to maintain the cell histone acetylation level. A significant increase of acetyl-CoA with 8C administration would diminish the need for Kat2a upregulation. Following this rationale, we discussed the change of Kat2a in line 381: “The change of Acetyl-CoA may also have a direct effect on the expression of HATs after I/R injury in heart. Kat2a, a major HAT enzyme in catalyzing histone acetylation, was upregulated after I/R, while enrichment of acetyl-CoA by 8C metabolism diminished the need of Kat2a upregulation.”

2) Echocardiography – Describe the device/machine used and indicate the anesthesia that was used if any.

We revised the methods describing echocardiography. “The rats were anesthetized with 2% isoflurane and echocardiography (ECG) was performed 1 day and 4 weeks after surgery using Vevo 2100 system with M model.”

3) What is the basis for estimated NRVM surface area as 6000μm3? This should be measured directly, or validation of this approach shown.

We estimate the NRVM sizes based on two previous studies described previously ^1, 2^. In the first paper, myocyte size was measured by histological sections. The size of mouse neonatal myocytes is ~5600 μm^3^ and that of the adult myocytes is 17,000 μm^3^. The size of the adult rat myocyte is about 18000 μm^3^. Therefore, we estimate that the NRVM size is ~6000 μm^3^. In the second paper, NRVM sizes are measured in culture with the 3 dimensions of NRVM at maximal 100 μm in length, 2-8 μm in thickness, and 10-40 μm in width. Based on these parameters, we estimate that the average size of NRVM is 75 (L) x 20 (D) x 4 (H) = 6000 μm^3^. We cited these two papers in the manuscript.

4) Numerous grammatical errors exist, and the manuscript should be editorially revised by an English language editor.

We thank the editor for pointing out the grammatical errors. We had the manuscript revised by a native English language editor.

References

1. Bensley JG, De Matteo R, Harding R and Black MJ. Three-dimensional direct measurement of cardiomyocyte volume, nuclearity, and ploidy in thick histological sections. *Sci Rep*. 2016;6:23756.

2. Yang D, Xi J, Xing Y, Tang X, Dai X, Li K, Li H, Lv X, Lu D and Wang H. A new method for neonatal rat ventricular myocyte purification using superparamagnetic iron oxide particles. *Int J Cardiol*. 2018;270:293-301.